# Study Analysis of Thermal, Dielectric, and Functional Characteristics of an Ethylene Polyethylene Diene Monomer Blended with End-of-Life Tire Microparticles Amounts

**DOI:** 10.3390/polym13040509

**Published:** 2021-02-08

**Authors:** Marc Marín-Genescà, Ramon Mujal-Rosas, Jordi García-Amorós, Miguel Mudarra, Xavier Ramis Juan, Xavier Colom Fajula

**Affiliations:** 1Mechanical Engineering Department, Escola Tècnica Superior d’Enginyeria Química, Rovira i Virgili University, 43007 Tarragona, Spain; 2Electrical Engineering Department, Escola d’Enginyeria de Terrassa, Technical University of Catalonia, 08222 Terrassa, Spain; mujal@ee.upc.edu; 3Electrical Engineering Department, Escola Tècnica Superior d’Enginyeria, Rovira i Virgili University, 43007 Tarragona, Spain; jordi.garcia-amoros@urv.cat; 4Physics Department, Universitat Politècnica de Catalunya, 08222 Terrassa, Spain; miguel.mudarra@upc.edu; 5Machines and Thermal Motors Department, Universitat Politècnica de Catalunya, 08222 Terrassa, Spain; xavier.ramis@upc.edu; 6Chemical Engineering Department, Escola d’Enginyeria de Terrassa, Technical University of Catalonia, 08222 Terrassa, Spain; xavier.colom@upc.edu

**Keywords:** dielectric properties, thermal properties, microstructure analysis, composites, recycling rubber

## Abstract

The recycling and disposal of disused tires is a topic of great concern to today’s companies, researchers, and society in general. In this sense, our research aims to recycle end-of-life tires (GTRs) through the separation of the fraction of vulcanized rubber from the other compounds in order to later grind this fraction and separate it into lower particle sizes. Finally, we aim to incorporate these GTR particles as a filler of an ethylene-polyethylene-diene monomer (EPDM). The obtained composites with EPDM and GTR are tested (5%, 10%, 20%) comparing these values with neat EPDM as a control sample. Thermal tests such as differential calorimetry (DSC) and thermogravimetric analysis (TGA) as well as dielectric tests (DEA) are performed in order to characterize these materials and check their viability as dielectric or semiconductor, for industrial use. It is checked how the presence of GTR increases functional properties such as conductivity/permittivity. The influence of temperature (40 to 120 °C) and addition of GTR particles in electrical properties has also been analyzed. The dielectric behavior of these composites is fully characterized, analyzing the different types of relaxation with increasing frequency (10 mHz to 3 MHz), using the electric modulus, and Argand diagrams among other measures. The influence of GTR and temperature in the dielectric and thermal behavior of these materials has been analyzed, where CB of GTR creates interfacial polarization phenomena in the dielectric behavior of the composite and increases the permittivity (real and imaginary) as well as the conductivity. Finally, with these obtained properties, the possible application of EPDM/GTR composites as industrial dielectrics has been studied.

## 1. Introduction

The recycling of tires once they have ended their use or useful life is one of the great problems posed to the environment [1,2] due to the great worldwide proliferation of this waste type, with a global rubber production of over 30 million tons yearly [1]. The tires finally end the useful lifetime, forming part of landfills, often uncontrolled or without any control measures, where they release their components and cause first-order pollution. This fact has special incidence when these landfills are in the oceans and seas due to the decomposition acceleration and pollution to the environment in a relevant way, further affecting flora and fauna.

The major obstruction to the recycling of tires is the presence of crosslinked rubber in the composition of the tire, which makes any attempt to recycle through reprocessing with a simple heat treatment completely impossible [3,4]. There are numerous processes for recycling and recovery of this waste; we will analyze some of the proposals in this regard: energy recovery, due to the use as a fuel of parts of tires or rubber. Another recycling issue is the devulcanization or decrosslinking of the structures remaining in the rubber. This solution is certainly the optimum, as it would enable recycling and it is possible to manufacture new tires from used tires. These methods have a certain advantage, that is, to make possible the complete reuse of the rubber, but they are difficult because they combine different types of treatments of different complexity (i.e., microwaves, ultrasounds, thermomechanical or biological treatments) [5,6,7,8,9,10,11]. Due to the complexity, these different devulcanize methods need an intermediate step before being able to reprocess again the rubber, and many times they do not achieve devulcanization to 100%. This means that the rubber reticulated structures have difficulties admitting other material in the inner; although, with previous treatments it is possible to put fractions of waste rubber materials without metallic and textile fraction in polymeric matrices. These obtained composites can have several applications such as in playgrounds, or as elements for public works, in urban parks, and others [12,13,14,15]. In this case, recycling is much easier than devulcanization because it only requires the crushing and separation of the particles of GTR from whole components of tires [16,17]. This type of recycling denotes a big simplicity that allowed to obtain a great number of commodity applications.

This research is based on the addition of GTR microparticles (diameter particle <200 µm) in a nonpolar elastomeric polymer such as EPDM (Ethylene Polyethylene Diene Monomer), nonvulcanized. Thus, in this research, we will assess the possible interaction of GTR particles in an EPDM polymeric matrix and then analyze both thermally and dielectrically behavior. In addition, the potential of these samples has been studied for possible applications based on these performed and analyzed composites (EPDM and GTR). In dielectric behavior, it is known that the incorporation of carbon black (CB) produces an increase of conductivity in several analyzed polymer composites [18,19,20,21,22,23,24], so it is expected that the presence of CB in tires increases this parameter and improves a conductive behavior. EPDM rubbers are good insulators with a relatively high electrical resistivity [25], with nonpolar chemical groups in their polymeric structure. However, the electrical properties of composites are more dependent on the additives used in the composite blends than on the matrix elastomeric base, so it defines the insulation properties for the low/medium voltage range.

Recent research has demonstrated the presence of relaxation of Maxwell-Wagner-Sillars (MWS) [26,27,28] in composites with tire particles in thermoplastic polymer matrices [29,30,31,32]. The present research aims to fully characterize the dielectric behavior completely for the GTR particles reinforcement EPDM composites and demonstrate the existence of such MWS relaxations in these materials and their possible use in dielectrics or also in semiconductor industrial applications. To analyze the dielectric behavior, the Electrical Module (M) will be used in these composite materials, both for neat EPDM without any reinforcement (0% GTR) and for samples with different percentages of GTR (5%, 10%, 20% GTR), to investigate the behavior that has these materials with the addition of particles of GTR.

## 2. Materials and Samples Processing

### 2.1. Materials and Samples Preparation

Ethylene Propylene Diene Rubber (EPDM) used as a composite basis material is an elastomeric polymer that has good abrasion resistance and wears. The composition of this material contains 70% ethylene, being, in general, more resistant the higher the percentage of ethylene. It is very easy to process, making, with extreme agility, parts by extrusion, molding, however, in its processing (Brabender mixer machine, Brabender GmbH & Co, Duisburg, Germany). The materials used can be easily processed with standard plastic mixing machines, such as the Brabender Machine type WPL50EHT, as in the present research manuscript. The used grinding recycled tire (GTR), with different particle sizes (<700 μm), was provided by GMN (Gestió Mediambiental de Neumàtics, SL, Maials, Spain) and it has been tested by TGA analysis that the carbon black content is around 30%. The original GTR particles were screened into particles <200 μm in diameter, by sieving. There has been performed four different types of samples materials for the present research: the first one, was performed using only EPDM (100% EPDM), the other samples have been added a range of ground tire rubber particles in different compositions (5%, 10%, and 20% of GTR). Ethylene Propylene Diene Monomer (EPDM) has been used without vulcanization, so any vulcanizing additive has been added to the formulation. The dienes generally range from 2.5% to up to 12% by weight. The qualities of this type of elastomer are its oxidation resistance. The incorporated diene in the polymerization phase makes it an elastomer that is difficult to oxidize, moreover its resistance to atmospheric agents is remarkable. Regarding heat resistance, the EPDM temperature range between −15 to 130 °C, and it can reach 140 °C (Table 1). This elastomer shows very notable resistance to water in general and good electrical resistance properties. About degradation, EPDM does not degrade or age easily and it is easy to process. EPDM elastomer used in the present research was acquired from Lanxess Chemicals, company, in pellets form, in 1–4 mm diameter granulated materials. The main characteristics of the EPDM used are in Table 1.

### 2.2. Compound Preparation Process

The GTR microparticles were selected employing a mechanical sieve with a diameter of less than 200 µm and dried in a forced-air oven at a temperature of 100 °C for 24 h. Subsequently, 4 samples of the composite material were prepared with GTR of compositions (0, 5%, 10%, 20%). The mixing process was done with a Brabender mixer-plastograph type W PL 50 EHT (Brabender GmbH & Co., Duisburg, Germany), at Processing Temperature (180 °C) to prevent polymer degradation, using a hot plate press in a Collin Mod. P 200 E (Dr. Collin GmbH, Germany). The laminates of the different base materials with GTR (150 mm × 150 mm × 2 mm) were obtained using a hot plate press at 200 bars, for 10 min. The cooling process was carried out with cold water under pressure until it reached room temperature, this process was carried out in the same press, under pressure, for 10 min. Samples were performed appropriately according to ASTM 150. A sample of the pure polymer (EPDM without GTR) was also prepared with the same requirements to obtain comparable results.

### 2.3. Test Methods

#### 2.3.1. Morphology Analysis

The morphology analysis was performed with Quanta 600 Scanning Electron Microscope (FEI company, Hillsboro, OR, USA), at a level voltage of 20 kV, with no sputtering, and 200 and 630 magnifications. The Quanta 600 FEG (FEI company, Hillsboro, OR, USA) is a field emission scanning electron microscope capable of generating and collecting high-resolution and low-vacuum images. The microphotography obtained has been used for morphology analysis, and specifically to analyze the dispersion of GTR particles in the EPDM matrix samples and to study the cohesion and interaction between composites phase: GTR (reinforcement) and EPDM (elastomeric matrix).

#### 2.3.2. Thermal Test Methods: TGA and DSC

The calorimetric analysis was performed using a Mettler DSC-822e calorimeter with a TSO801RO robotic arm, calibrated using an indium (heat flow calibration and temperature calibration) and zinc (temperature calibration) standards. Samples of approximately 10 mg of the mass were deposited in the aluminum pans under a nitrogen atmosphere to test performance from −75 to 150 °C at 10 °C/min. DSC was used to determine glass transition temperatures (*T*_g_) and melting data. *T*_g_’s were determined as the temperature of the half-way point of the jump in the heat capacity when the material changed from a glassy to a rubbery state and the error was estimated as ±1 °C. Thermogravimetric analysis (TGA, Mettler Toledo, Columbus, OH, USA) was carried out with a Mettler TGA/SDTA 851e/LF/1100 thermobalance. Samples with an approximate mass of 10 mg were decomposed between 30 and 800 °C at a heating rate of 10 °C/min in N2 atmosphere (50 cm^3^/min measured in normal conditions). TGA was used to study the thermal stability of the formulations and to differentiate the different components of the formulations.

#### 2.3.3. Dielectric Test Methods: DEA

The test specimens for electrical tests (ASTM D-150 [33]) are cylindrical with 25 mm diameter and 0.1 mm thickness. To perform the electrical tests, the DEA (Dynamic Electric Analysis, Novocontrol Technologies GmbH & Co, Montabaur, Germany) spectroscopic technique was used, introducing the assembly into a test chamber, whose function is to provide the required temperature. The parameters and dielectric magnitudes were measured through DEA dielectric test (Dielectric Analysis) [34,35] with a BDS40 by Novocontrol. Temperature control was carried out with a Novotherm system. Golden plated parallel electrodes with 2 cm diameter were used. Samples thickness was 1 mm. The measures were undertaken on a range of frequencies from 1 mHz and 3 MHz, and on 10 °C steps from 30 to 120 °C. Complex permittivity, modulus, and conductivity are calculated from the complex impedance tested.

## 3. Thermal and Morphological Results

### 3.1. Microphotography’s: Morphology Analysis

Figure 1 shows the complete dispersion of GTR particles in the EPDM elastomeric matrix in the sample of 20% GTR (Figure 1a,b) and 10% GTR (Figure 1c,d) with different magnification (200× and 630×). The dispersion analyzed with the images at 200 magnifications (Figure 1a,c) shows a suitable dispersion of particles in the sample, showing a homogeneous distribution of GTR in samples. On the other hand, Figure 1a,d shows a low interaction between GTR and EPDM elastomeric matrix. It can be explained due to lack of compatibility between both phases, therefore we can conclude that the two phases (GTR and EPDM) have a rather low affinity that explains the low interaction between GTR particles and the matrix, as can be described from the analyzed images.

### 3.2. Thermal Properties

#### 3.2.1. Thermogravimetric Analysis (TGA)

Figure 2a shows the thermogravimetric curves for all formulations studied. It can be observed that the degradation took place in two steps except for a neat EPDM. These degradation steps can be easily assigned to the constituent parts because the second step appeared at the same temperature as the neat sample of 100% EPDM, in Figure 2b, observe the shoulders associated to GTR in the negative peaks of all non-neat formulations. The first decomposition step takes place at the temperature range of 350–400 °C, corresponds to the degradation of the GTR and the second degradation step occurring at the temperature range 450–500 °C can be attributed to the thermal decomposition of the EPDM. The weight loss associated with each degradation step correlates with the weight composition of samples. Figure 2 shows the TGA of EPDM/GTR samples, and it can observe that the thermogram moves to a lower temperature whit increasing GTR. This means that the sample is less stable and initiates the decomposition at a lower temperature. The initial decomposition temperature of the 80% EPDM decreases practically 100 °C, and it falls in the range of temperatures reported for reworkable materials. This result can be useful when it is necessary to decompose under controlled conditions to recover, for example, a substrate with higher added value, when EPDM/GTR composites are used as a coating.

In Figure 2b is checked the GTR degradation in the heating up to 500 °C which corresponds to the degradation of elastomers. Two minimums can be seen in GTR curve which means that there are two different elastomers in the GTR composition, the first one is natural rubber and the second is styrene-butadiene.

#### 3.2.2. Differential Scanning Calorimetry (DSC)

The polymeric materials are usually characterized by two main types of transition temperatures: the crystalline melting temperature Tm (or crystalline melting point) and the glass transition temperature *T*_g_. The crystalline melting temperature is the melting temperature of the crystalline domains of a polymer sample. The glass transition temperature is the temperature at which the amorphous domains of a polymer changes from the glassy to the rubbery state. Semicrystalline polymers, such as EPDM, show a kind of transition. Tm is a first-order transition with a discontinuous change in the specific volume at the transition temperature. *T*_g_ is a second-order transition that involves only a change in the temperature coefficient of the specific volume. The two thermal transitions are generally affected in the same way by the molecular symmetry, the structural rigidity, and the secondary attractive forces of the polymer chains. By DSC, *T*_g_ is shown as a change in heat capacity and Tm as an endothermic peak.

Figure 3 shows the DSC thermograms at 10 °C/min of al formulations studied. It can be observed the *T*_g_ (glass transition) at low temperature (amorphous part) and the melting peak (crystalline part) at a higher temperature. Ethylene-Propylene-Diene-Monomer Rubber (EPDM) shows a semicrystalline behavior with values of *T*_g_ and *T*_m_ (Table 2) similar to the reported in the literature [36].

All the material samples analyzed give similar results. In this sense, the *T*_g_ and *T*_m_ hardly change with the GTR amount (Figure 3 and Table 2), which indicates the low interactions between EPDMA and GTR. Enthalpy of fusion slightly decreases on increasing GTR content according to the relative amount of both components in the formulation and to the amorphous character of GTR.

## 4. Dielectric Test Results

The dielectric response of polymeric materials at various frequencies can be described in terms of the complex relative permittivity *ε_r_**. To describe the effect of the polarization of the electrode and solve the relaxation at low frequencies, the term “electric modulus” was used. The electric modulus was introduced by McCrum et al. [37] and is used for the study of electrical relaxation phenomena in many polymers [38,39,40,41] and has been used for the present dielectric characterization and analysis. Electrical Modulus is defined by Equation (1), where *M*′ and *M*″ are, the real and imaginary part of the electrical modulus.
(1)$$M_{r}^{*}=\frac{1}{\varepsilon_{r}^{*}}=\frac{\varepsilon_{r}^{\prime }}{\varepsilon_{r}^{\prime }-j\varepsilon_{r}^{\prime \prime }}+j\frac{\varepsilon_{r}^{\prime \prime }}{\varepsilon_{r}^{\prime 2}+\varepsilon_{r}^{\prime \prime 2}}=M_{r}^{\prime }+jM_{r}^{\prime \prime }$$

From the physical point of view, the electric modulus corresponds to the relaxation of the electric field in the material when the electric displacement remains constant so that the electric modulus represents the real dielectric relaxation process. Despite the fact that there is no additional information in the electric modulus, in the case of the conductive processes that are observed at low frequencies, the loss factor exhibits a sharp increase whereas the imaginary part of the electric modulus shows a peak, so that this function is suitable to study the space charge relaxation phenomena, as they are reflected by the changes of this peak.

### 4.1. Permittivity Analysis

#### 4.1.1. Real Permittivity (*ε*′) or Dielectric Constant

Figure 4 and Figure 5 show the values of the real permittivity and imaginary permittivity in the range of the frequency for different temperatures (from 40 to 120 °C) for each one of EPDM/GTR composites. According to Figure 4 and Figure 5 both, real and imaginary permittivity are influenced by the content of GTR, frequency, and temperature. Both increase with increasing GTR content and increases also with temperature, at the same level range of frequencies, and decreases of *ε*′ with rises of frequency, at the same temperature and/or GTR percent. In this sense, increases in frequency cause slight increases in losses (*ε*″) in compounds with low GTR contents until reaching a maximum, after which the effect is slightly reversed. As the temperature increases, the permittivity increases because the mobility of some segments in the polymer chain also increases. Although in general, the shape of the real part of the permittivity undergoes minor changes.

#### 4.1.2. Imaginary Permittivity (*ε*″)

A similar trend can be observed for imaginary permittivity (*ε*″) in Figure 5. The imaginary permittivity increases with the amount of GTR amount on EPDM composites, in the low, medium, and high frequencies, and with the increase of temperature, except in the samples of neat EPDM just the opposite temperature dependence relation (the higher temperature the lower *ε*″). Another clear trend observed in Figure 5 is the relaxation peak of permittivity shift of the peak relaxation frequency to higher frequency values with temperature rises. It can identify two effects from GTR amount in the samples: direct effects and indirect effects. About direct effects (red arrow in Figure 5) it can check rises of *ε*″ with the growth of GTR amounts in the samples at low frequencies and rise of the peak frequency of relaxation at middle frequencies with ascent temperatures. About indirect effect, frequencies (green circle) are checked the rises on *ε*″ relaxation peak (over 10^1^–10^3^ Hz) and a distortion (over 10^4^ Hz).

### 4.2. Conductivity Analysis: (DC/AC) Regime

At constant temperature, the conductivity AC can be expressed as σ_ac_ = σ_dc_ + A·ω^s^, where σ_dc_ is the limit value (ω → 0) of σ_ac_, A and s are parameters dependent on temperature and filler content [42,43], therefore, the AC conductivity (σ_ac_ = *ε*_0_ · ω · *ε*″) of composite materials depends on the test temperature and the frequency ranges, and conductivity rises with increasing one of them. This can be seen in Figure 6, where at low frequencies the influence of DC conduction (direct current) is very important and the dependence on the frequency can be expressed through a power law of the type σ_ac_ ~ ω_s_ (0 < s < 1) [44,45]. In general, conductivity is higher in samples with higher GTR particle content, which suggests that GTR favors the transfer of electrical charges at the EPDM-GTR composite interface. The conductivity values alternate by an order of magnitude with temperature, indicating the existence of thermally activated processes.

### 4.3. Electrical Modulus

#### 4.3.1. Real Electrical Modulus

To facilitate the study of relaxation phenomena, the electrical modulus formalism is commonly used [46,47]. In Figure 7 and Figure 8 the real electrical modulus (*M*′) and imaginary electrical modulus (*M*″) are observed for compounds with different concentrations of GTR as a function of the frequency and in a temperature range from 40 to 120 °C. In the composite samples, interfacial polarization phenomena are happening due to the presence of the GTR phase in the EPDM and GTR composite. *M*′ rises with ascents frequency up to reach a constant value, over 10^−6^ Hz. In general, low changes are seen in *M*′ (Figure 8) with percent GTR of increasing amounts.

#### 4.3.2. Imaginary Modulus

In turn, in the same frequency interval, a peak in *M*″ develops, indicating the presence of relaxation processes. These peaks correspond to α relaxations that are formed at high temperatures (≈*T*_g_ of the resin) when the mobility of the polymer molecules is high enough [48]. On the other hand, unlike what occurs in the measurement of dielectric losses, in the case of the study of *M*″ a new relaxation at high frequencies can be appreciated that can be attributed to a phenomenon of the Maxwell-Wagner-Sillars type (MWS). This kind of relaxation has its origin in the interfacial polarization and appears in heterogeneous materials, such as partially crystalline or semicrystalline polymers, such as EPDM, they contain crystalline regions with conductivity and permittivity different from the amorphous regions [49]. Therefore, Figure 9 allows us to simultaneously see the evolution with temperature and frequency for different percent amount of GTR in the EPDM matrix. At least three relaxation peaks are observed: β (low frequencies, below 10^−2^ Hz), α (medium frequencies, over 10^2^ Hz), and MWS (medium/high frequencies, over 10^4^ Hz). These dielectric relaxation peaks are thermally activated phenomena, and they are moved to higher frequencies as the temperature increases as we have seen in Figure 9. This allows an analysis of the dependence of each dielectric relaxation with temperature.

At the present analysis, it appears a second relaxation peak with the presence of GTR at high frequencies, the imaginary component of the dielectric module presents a relaxation that can be attributed to a phenomenon of the Maxwell-Wagner-Sillars type. Due to ionic conduction effects at medium/high frequencies, that peak is not directly detectable in the spectrum of permittivity [50]. This class of peak relaxation has its origin in interfacial polarization and appears in heterogeneous materials, like GTR composites [51]. However, because semicrystalline polymers like such as PP [52] or EPDM contain crystalline regions with conductivity and permittivity different from amorphous regions, it turns out that they are also able to detect this interfacial polarization peak [53].

### 4.4. Argand Diagram

Argand diagram analysis does not detect any semicircles that lead to Debye-like behavior [54], so neither neat EPDM nor EPDM/GTR samples have non-Debye behavior in any of the cases analyzed. It has not been observed that a Coelho relaxation [55] can be reproduced that can be deduced from the Argand diagram because the semicircle characteristic of this relaxation is not observed in Figure 9. In the Argand diagram (Figure 9) it is observed that dielectric relaxations take place from *M*′ approximately at 0.5–0.9, this dielectric relaxation is of alpha type. Thermally we see how the alpha relaxation is activated in an obvious way, more than the other two observed relaxations. Besides, it is observed from 10% and 20% how the initial relaxation is decomposed into 2 relaxations: MWS and α (Figure 9c).

### 4.5. Relaxations Type (α, β, MWS) and Activation Energy (Ea)

At high frequencies, an *M*″ and *ε*″ peak can be seen that can be associated with the MWS relaxation of EPDM (MWS Peak relaxation), as are seen in Figure 8c,d and Figure 9c,d. The behavior of this relaxation, which is linked to interfacial polarization, is due to the expression of Arrhenius for thermally activated relaxations, according to the following Equation (2):
(2)$$f=f_{0}e^{-\frac{E_{a}}{kT}}$$
where *f* is the maximum frequency of relaxation, *f*_0_ is the natural frequency (at T = ∞), *E_a_* is the activation energy, k is the Boltzmann constant, and T is the temperature in degrees Kelvin. Activation energies of between 0.44 eV and 0.35 eV were obtained in the adjustments made with the EPDM/GTR samples (Figure 10). However, observing a decreasing trend in *E_a_* with increasing GTR concentration (10 to 20% GTR), according to Table 3. This trend is logical because, with incorporation of GTR, less energy is required to energetic activation of GTR-EPDM mixtures analyzed, due to the capacitive effect that occurs at the GTR-EPDM interface with the accumulation of opposite sign charges in the interface composite GTR-EPDM, called interfacial polarization process. The increasing percentages of GTR shows a Maxwell-Wagner-Sillars (MWS) relaxation type, observed mainly in the analyzed 10%–20% GTR samples. This prominent dielectric relaxation is seen in Figure 8 and Figure 9 [56,57]. On the other hand, we should also note the visualization of the beginning of a dielectric relaxation at very low frequencies (<10^−2^ Hz) and which can be identified as a relaxation of β type.

## 5. Discussion: GTR Effect and Composites Application

The value of the dielectric constant is closely related to the polarization orientation. When the orientation increases, the dielectric constant increases as well. Polarization orientation is due to the presence of permanent dipoles in a molecule, so, logically, changes in a structure created by the presence of GTR create increases in the dipole moments, and therefore also in the polarization and dielectric constant. On the other hand, analysis of Figure 6 on conductivity characteristics shows a shift towards higher frequencies observed when imaginary conductivity appears, with the increase of GTR in the analyzed samples. In this sense, this rises in GTR amounts also ascents the imaginary conductivity of the analyzed samples; this is consistent with interfacial polarization phenomena, such as those we tested in the EPDM and GTR samples.

According to electrical criteria, a suitable application for spacers for power lines used in electrical installations to separate the electrical cables from others or other parts from installation, according to International Electrotechnical Committee (IEC) 61854, is as follows: overhead lines—requirements and tests for spacers, the value of resistivity needs to be >5.5 × 10^5^ Ω·cm or <1.8 × 10^−5^ S/cm, so the mixtures of EPDM and GTR from 5% to 20%GTR (Table 4) accomplish this criterion of conductivity is suitable for the aforementioned application [58,59]. Flexible cable spacers are electrical accessories designed to provide uniform spacing between wire conductors in a bundle of two horizontal conductors, and they are used to ensure electrical characteristics of the cables and to minimize vibration and thermal effects, avoiding the appearance of damage to the conductors.

## 6. Conclusions

At the morphological level, we can conclude that the analysis of microphotographs of the analyzed GTR and EPDM composites shows a good dispersion of the particles in the composite EDPM but a low cohesion of the reinforcing particles with the polymer matrix. This fact is also confirmed by DSC that hardly shows GTR content influence on the thermal transitions, in the thermal parameters, attributable to the low interaction between both components of composites.

At the dielectric level, electrical properties observed are logical increases in conductivity due to the conductive fraction of carbon black that forms (35%) of GTR particles which configure the composite reinforcement. One of the most notable effects of the presence of GTR fractions in the EPDM matrix is the appearance, from 5% and up to 20% of GTR of highly visible interfacial polarization phenomena, in the form of a Maxwell-Wagner-Sillars (MWS) relaxation, this relaxation appears in composite materials when an interface appears, similar than GTR composites analyzed. In the present experience, this MWS relaxation appears almost after the medium frequency relaxation of the β EPDM relaxation, identified. This fact is explained by many opposite signs appearing at the interface between the GTR reinforcement and the EPDM matrix (interphase).

Activation energies have been analyzed for MWS relaxations identified in the permittivity analysis, analysis of electrical modules as well as with Argand diagrams. These activation energies are reduced with the increasing presence of GTR, which is explained by the fact that with the presence of GTR, less energy will be required to activate the polarization processes that cause MWS relaxations in the analyzed composites.

The addition of GTR to the EPDM matrix induces changes in the dipole moments of the material, modifying and creating interfacial polarization phenomena in the dielectric behavior of the composite and increases the permittivity (real and imaginary) as well as the conductivity due to the presence of carbon black coming from GTR placed between the chains of the matrix polymer, which increases the conductive properties of the composite. However, these increases do not experience large variations because EPDM is a nonpolar polymer. Broadly, one of the main conclusions is that the incorporation of increasing amounts of GTR into a polymeric EPDM matrix does not significantly increase the conductivity of the material even though it has a good amount of CB (30%). One explanation is that the EPDM nonpolarity attenuates the conductive behavior of GTR in an EPDM composite for fractions up to 20% of GTR. Moreover, EPDM and GTR composites properly maintain thermal and dielectric properties, so the application of EPDM loaded with GTR fractions could be used in a low dielectric requirements application like spacers of cable and electrical lines, according to conductivity data obtained from the dielectric test. This gives an alternative recycling GTR method instead of other complex rubber-devulcanization methods.

## Figures and Tables

**Figure 1 polymers-13-00509-f001:**
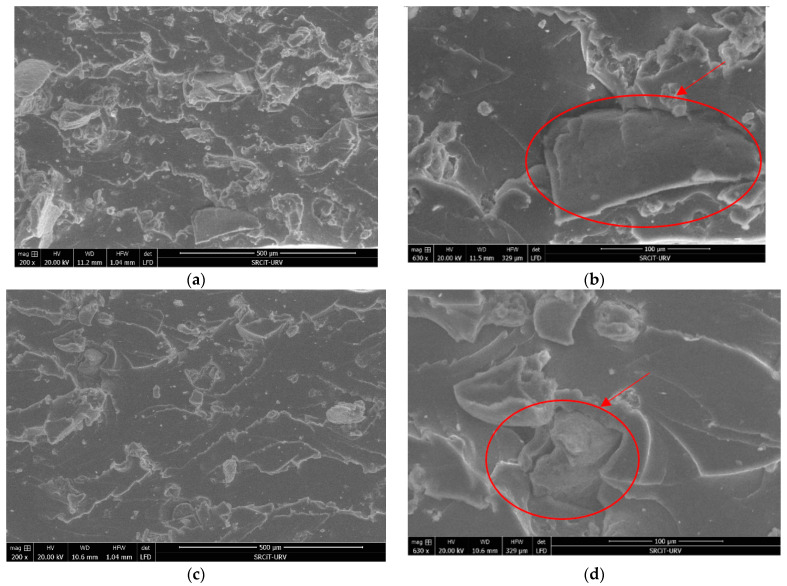
SEM micrographs of EDPM at 20% GTR (**a**,**b**) and at 10% GTR (**c**,**d**), and at different magnifications: 200× (**a**,**c**) and 630× (**b**,**d**).

**Figure 2 polymers-13-00509-f002:**
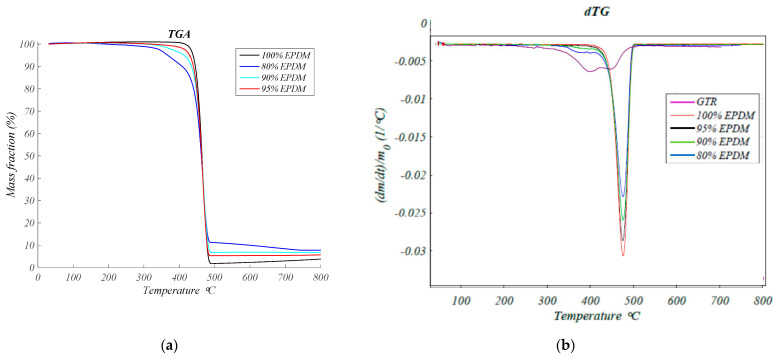
Thermogravimetric analysis: (**a**) Mass fraction (%) versus Temperature (°C), for EPDM and EPDM/GTR samples; (**b**) Rate of weight loss versus Temperature (°C), for EPDM and EPDM/GTR samples.

**Figure 3 polymers-13-00509-f003:**
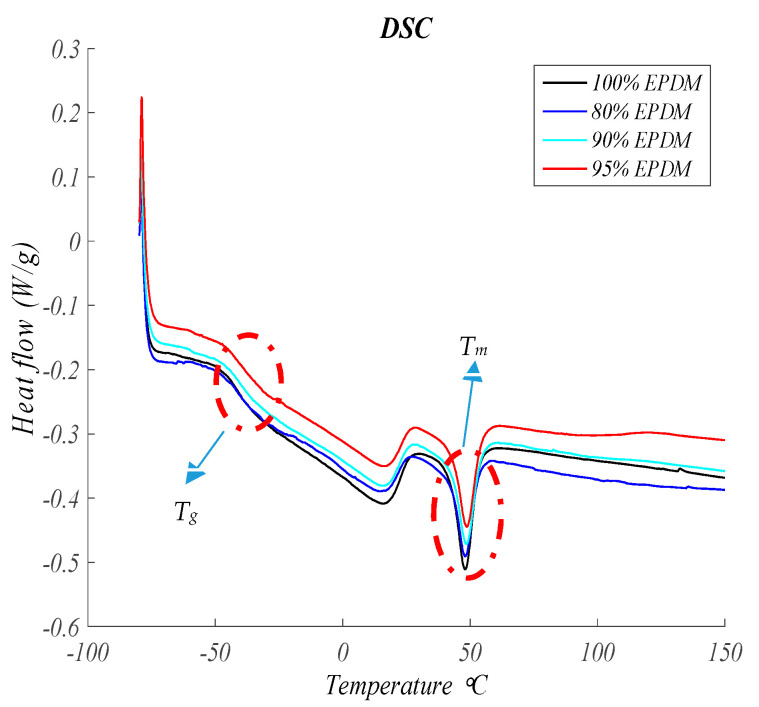
Heat flow versus Temperature at 10 °C/min for EPDM and EPDM and GTR samples.

**Figure 4 polymers-13-00509-f004:**
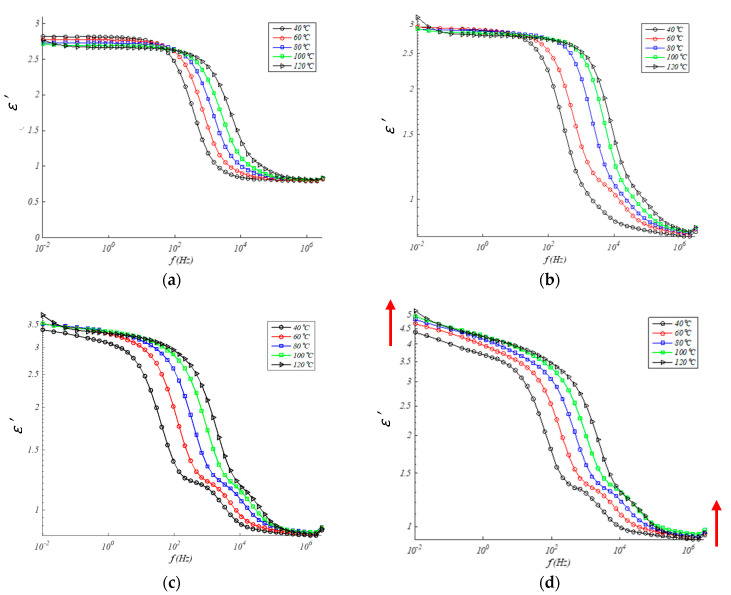
Real permittivity in EPDM composites for: (**a**) 0% GTR or neat EPDM; (**b**) EPDM + 5% GTR; (**c**) EPDM + 10% GTR; (**d**) EPDM + 20% GTR.

**Figure 5 polymers-13-00509-f005:**
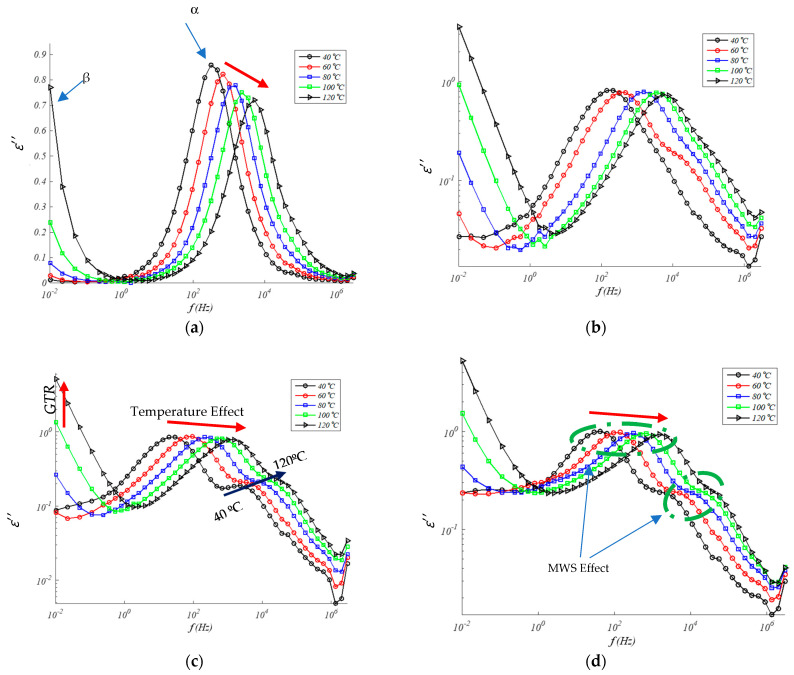
Imaginary permittivity in EPDM composites for: (**a**) 0% GTR or neat EPDM; (**b**) EPDM + 5% GTR; (**c**) EPDM + 10% GTR; (**d**) EPDM + 20% GTR.

**Figure 6 polymers-13-00509-f006:**
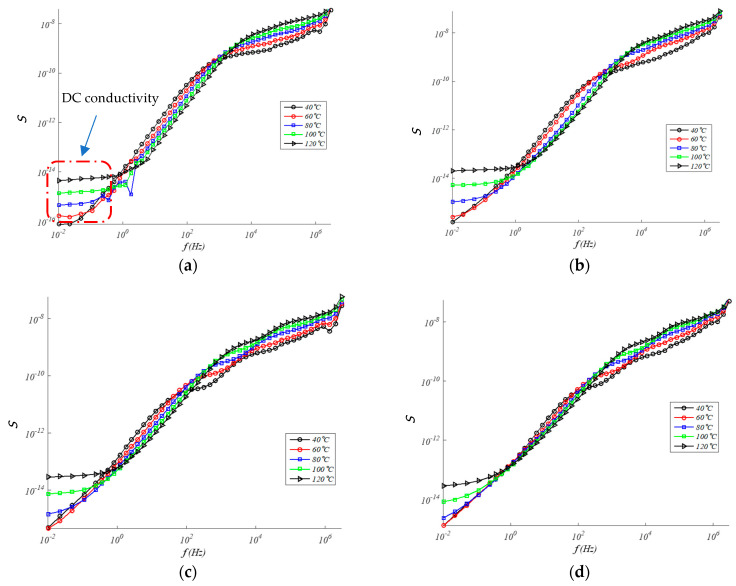
S-Conductivity (S/cm) in EPDM composites for: (**a**) 0% GTR or neat EPDM; (**b**) EPDM + 5% GTR; (**c**) EPDM + 10% GTR; (**d**) EPDM + 20% GTR.

**Figure 7 polymers-13-00509-f007:**
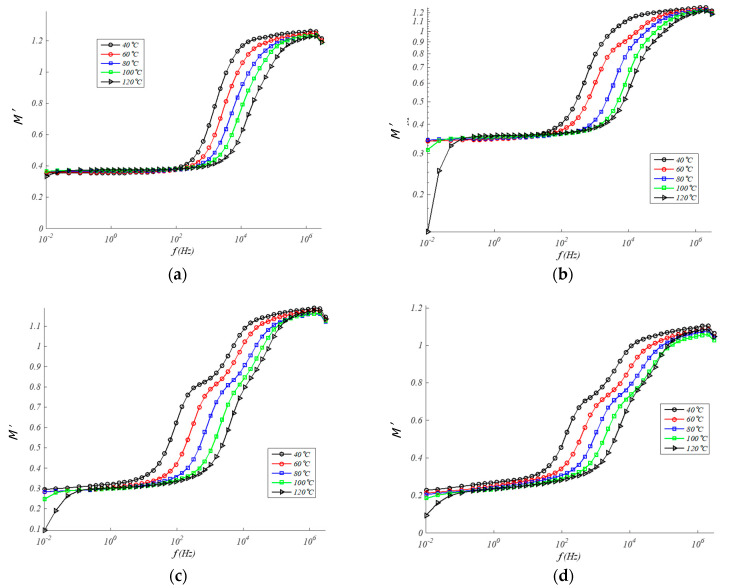
Real modulus in EPDM composites for: (**a**) 0% GTR or neat EPDM; (**b**) EPDM + 5% GTR; (**c**) EPDM + 10% GTR; (**d**) EPDM + 20% GTR.

**Figure 8 polymers-13-00509-f008:**
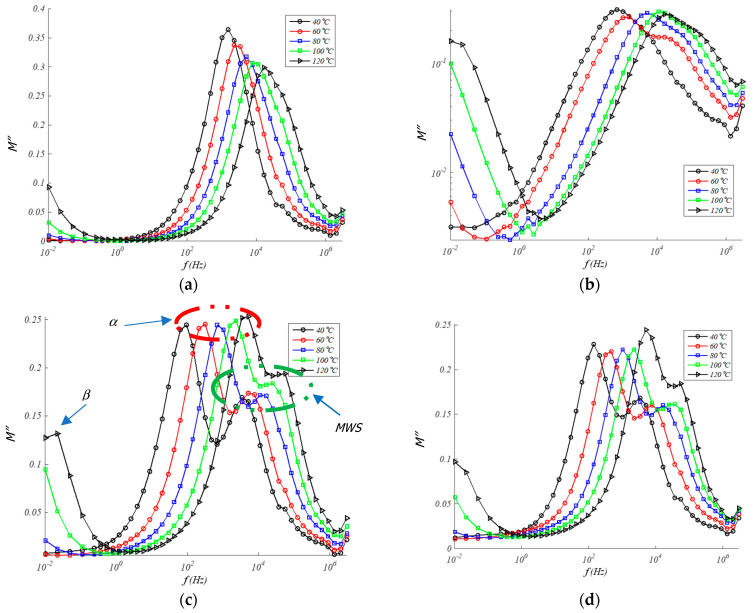
Imaginary modulus in EPDM composites for: (**a**) 0% GTR or neat EPDM; (**b**) EPDM + 5% GTR; (**c**) EPDM + 10% GTR; (**d**) EPDM + 20% GTR.

**Figure 9 polymers-13-00509-f009:**
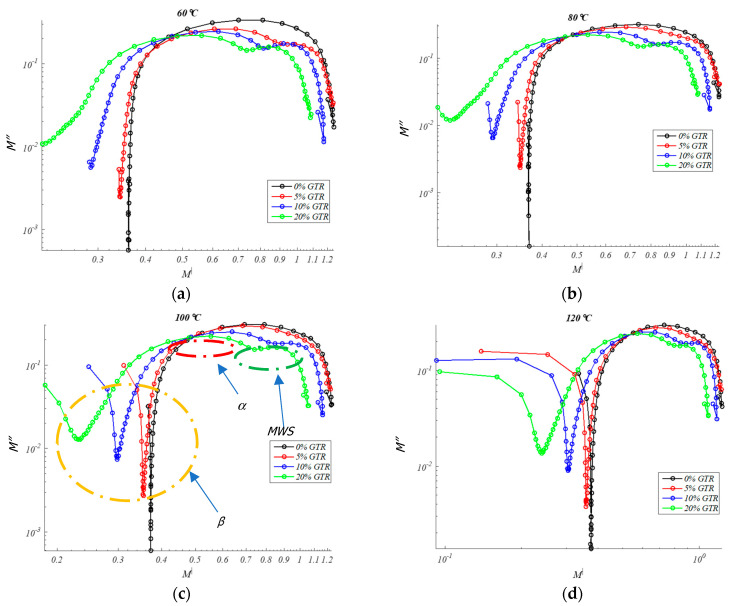
Argand diagram (*M*″-*M*′) in EPDM/GTR composites: for 0%, 5%, 10%, 20% GTR amount, for: (**a**) 60 °C; (**b**) 80 °C; (**c**) 100 °C; (**d**) 120 °C.

**Figure 10 polymers-13-00509-f010:**
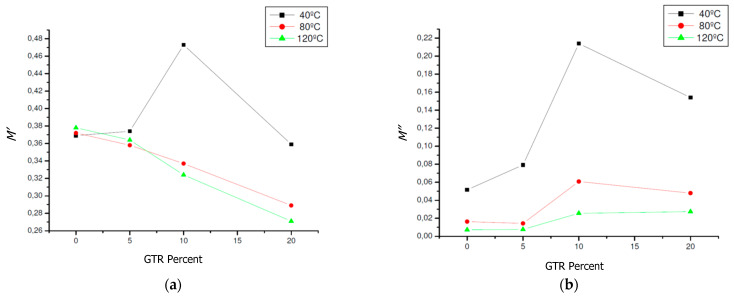
EPDM and GTR samples composites dielectric behavior, at frequency: 50 Hz, and different temperatures: 40 °C, 80 °C and 120 °C (**a**) real modulus; (**b**) imaginary modulus; (**c**) real permittivity; (**d**) imaginary permittivity; (**e**) S-Conductivity (S/cm) in EPDM composites for neat EPDM (0% GTR) and EPDM and GTR (5–10–20% GTR).

**Table 1 polymers-13-00509-t001:** EPDM Features.

**Mechanical Properties**	
Shore Hardness, Shore A	40–90
Tensile strength	25 MPa
Density	0.90–2.00 g·cm^−^³
**EPDM Physical Characteristics** (by manufacturer)	
Hardness	60° Shore A
Specific weight	1.55 kg/dm^3^
Apparent density	1.56 kg/L
Elongation at break	750%
Stretch resistance	6 N/mm^2^
Granulated volume	0.73 kg/L

**Table 2 polymers-13-00509-t002:** Glass transition temperature (*T*_g_), melting peak (*T*_m_), and enthalpy of fusion for EPDM and GTR + EPDM composites.

*Composition*	*Glass Transition Temperature* *T_g_ (°C)*	*Melting Peak* *T_m_ (°C)*	*Enthalpy of Fusion* *(J/g)*
100% EPDM	−41.88 °C	48.00 °C	9.06 J/g
95% EPDM	−39.39 °C	48.67 °C	8.54 J/g
90% EPDM	−40.46 °C	48.50 °C	8.55 J/g
80% EPDM	−39.73 °C	48.00 °C	7.68 J/g

**Table 3 polymers-13-00509-t003:** Evolution of Peak β and Maxwell-Wagner-Sillars (MWS) Relaxations, for EPDM composites with 10 and 20% GTR amounts in EPDM composites, at 50 Hz, and evolution of activation energies (*E_a_*) for 10 and 20% of GTR contents in EPDM composites.

% GTR	Temperature (°C)	α Peak Relaxation (Freq.; Hz)	α Peak Relaxation (*M*″)	MWS Relaxation (Freq.; Hz)	MWS Relaxation (*M*″)
10	40	9.52 × 10^1^	2.44 × 10^−1^	3.43 × 10^3^	1.69 × 10^−1^
	60	3.14 × 10^2^	2.45 × 10^−1^	5.11 × 10^3^	1.73 × 10^−1^
	80	6.98 × 10^2^	2.44 × 10^−1^	1.13 × 10^4^	1.71 × 10^−1^
	100	7.61 × 10^3^	3.07 × 10^−1^	8.31 × 10^4^	1.70 × 10^−1^
	120	5.11 × 10^3^	2.53 × 10^−1^	5.58 × 10^4^	1.94 × 10^−1^
20	40	1.41 × 10^2^	2.21 × 10^−1^	3.43 × 10^3^	1.73 × 10^−1^
	60	4.68 × 10^2^	2.20 × 10^−1^	7.61 × 10^3^	1.60 × 10^−1^
	80	1.03 × 10^3^	2.22 × 10^−1^	1.69 × 10^4^	1.59 × 10^−1^
	100	2.30 × 10^3^	2.22 × 10^−1^	2.51 × 10^4^	1.61 × 10^−1^
	120	5.11 × 10^3^	2.44 × 10^−1^	5.58 × 10^4^	1.84 × 10^−1^
**Activation Energy (*Ea*)**					
**%GTR**	**MWS Peak**				
10	0.4429 eV				
20	0.3593 eV				

**Table 4 polymers-13-00509-t004:** Evolution of real modulus; imaginary modulus; real permittivity; imaginary permittivity and conductivity (S/cm), for EPDM composites with GTR growing amounts, at 50 Hz and 40, 80 and 120 °C.

% GTR	Eps’	Eps’’	Modulus’	Modulus’’	S [S/cm]	Temperature
0	2.66	3.72 × 10^−1^	3.68 × 10^−1^	5.15 × 10^−2^	8.88 × 10^−12^	40 °C
5	2.55	5.41 × 10^−1^	3.73 × 10^−1^	7.91 × 10^−2^	1.29 × 10^−11^	
10	1.75	7.95 × 10^−1^	4.72 × 10^−1^	2.14 × 10^−1^	1.89 × 10^−11^	
20	2.35	1.00	3.58 × 10^−1^	1.53 × 10^−1^	2.40 × 10^−11^	
0	2.68	1.17 × 10^−1^	3.72 × 10^−1^	1.62 × 10^−2^	2.79 × 10^−12^	80 °C
5	2.78	1.11 × 10^−1^	3.58 × 10^−1^	1.42 × 10^−2^	2.65 × 10^−12^	
10	2.87	5.19 × 10^−1^	3.36 × 10^−1^	6.08 × 10^−2^	1.23 × 10^−11^	
20	3.36	5.57 × 10^−1^	2.89 × 10^−1^	4.79 × 10^−2^	1.33 × 10^−11^	
0	2.64	5.00 × 10^−1^	3.78 × 10^−1^	7.15 × 10^−3^	1.19 × 10^−12^	120 °C
5	2.74	5.77 × 10^−1^	3.63 × 10^−1^	7.63 × 10^−3^	1.37 × 10^−12^	
10	3.06	2.41 × 10^−1^	3.24 × 10^−1^	2.55 × 10^−2^	5.75 × 10^−12^	
20	3.65	3.69 × 10^−1^	2.70 × 10^−1^	2.73 × 10^−2^	8.81 × 10^−12^	

## Data Availability

Not applicable.

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
