# Peer review of "Study Analysis of Thermal, Dielectric, and Functional Characteristics of an Ethylene Polyethylene Diene Monomer Blended with End-of-Life Tire Microparticles Amounts"

_polymers, 2021, doi:10.3390/polym13040509_

Round 1

Reviewer 1 Report

The paper presents a study according analysis of thermal and dielectric properties of Ethylene Polyethylene Diene Monomer blended with end of life tire micro-particles amounts. Authors say, that recycling and disposal of tires is nowadays one of the great topics of the companies and researchers. Authors research aims to recycle end of life tires (GTRs) through the separation of the fraction of vulcanized rubber of the other compounds, to later grind this fraction and separate into lower particle sizes. Obtained by authors composites with EPDM+GTR were tested comparing these values with neat EPDM as a control sample. For example thermal tests and thermo-gravimetric (TGA) analysis and dielectric tests (DEA) were performed to characterize materials and verify viability as dielectric or semiconductor, for industrial use. it has been studied the possible application of EPDM/GTR composites as industrial dielectrics.

Comments and discussions:

  1. First chapter characterizes recycling of tires. Authors explain the volume of tires and methods their use in industry. They indicate areas where used tires can be used. I think, the chapter is well organized and written, used number of references is correct.
  2. Next chapter explain used materials and sample processing. Authors present materials and samples preparation. They explain material properties, such as mechanical and physical properties. Next they show compound preparation process. Authors present test methods, and morphology analysis. They explain thermal test Methods, such as TGA and DSC. Authors show dielectric test methods, means DEA.
  3. Chapter 3 shows obtained thermal and morphological results, such as microphotography, and SEM micrographs of EDPM. Next, they present thermal properties, such as thermos-gravimetric analysis (TGA), and differential scanning calorimetry (DSC). Authors indicate glass transition temperature, melting peak, and enthalpy of fusion for EPDM and GTR+EPDM composites.
  4. Next chapter shows dielectric test results. Authors present permittivity analysis, such as real permittivity and imaginary one – tan(delta). Next, they explain conductivity behavior of analyzed materials. Authors present electrical modulus, such as real electrical modulus, and imaginary one. They show Argand diagram, and relaxations type (α, β, MWS) and activation energy. In order to complete, I would measure also electrical strength, means breakdown voltage, but it is not necessary.

Author Response

REVIEWER 1

Comments and discussions:

  1. First chapter characterizes recycling of tires. Authors explain the volume of tires and methods their use in industry. They indicate areas where used tires can be used. I think, the chapter is well organized and written, used number of references is correct.

Answer: thank you for your comment.

  1. Next chapter explain used materials and sample processing. Authors present materials and samples preparation. They explain material properties, such as mechanical and physical properties. Next they show compound preparation process. Authors present test methods, and morphology analysis. They explain thermal test Methods, such as TGA and DSC. Authors show dielectric test methods, means DEA.

Answer: thanks to the reviewer for the comment.

  1. Chapter 3 shows obtained thermal and morphological results, such as microphotography, and SEM micrographs of EDPM. Next, they present thermal properties, such as thermos-gravimetric analysis (TGA), and differential scanning calorimetry (DSC). Authors indicate glass transition temperature, melting peak, and enthalpy of fusion for EPDM and GTR+EPDM composites.

Answer: thank you very much for your comment.

  1. Next chapter shows dielectric test results. Authors present permittivity analysis, such as real permittivity and imaginary one – tan(delta). Next, they explain conductivity behavior of analyzed materials. Authors present electrical modulus, such as real electrical modulus, and imaginary one. They show Argand diagram, and relaxations type (α, β, MWS) and activation energy. In order to complete, I would measure also electrical strength, means breakdown voltage, but it is not necessary.

Answer: Thank the reviewer for your comments and suggestions. Electrical strength, like breakdown voltage test, will be performed in further research, our research group will provide in next works manuscript a specific study-analysis from GTR influence on breakdown strength properties in a polymeric blend, providing a deep statistical study like demand this type of tests.

Reviewer 2 Report

- In the literature published so far, there are many reports on composites' production with a matrix from various thermoplastic polymers filled with GTR. Many of them also describe the dielectric properties of such copies, defining the final products as potential insulators. Earlier works by the authors also present very similar studies. Considering that the dielectric properties of EPDM are known, the final result of the research is predictable. Therefore, in my opinion, the novelty of the presented work raises doubts.

- Apart from the lack of a significant aspect of novelty, the work constitutes a complete elaboration of the topic and describes the produced materials' properties clearly. The experimental work plan is prepared correctly, and the descriptions of the research results are reliable and well-justified with references.

- The work requires mainly editing corrections related to the presentation of measurement data. Correction of the diagrams (e.g. Fig. 9) should be taken into account by presenting the test results on the same scale and correcting the descriptions under the diagrams. The axes in other drawings and tables must also be corrected by standardizing marking decimal places (I suggest resigning from engineering designations, e.g. E-11) and repeating descriptions under the drawings.

- All the drawings must be redrawn using the same size and type of fonts. Moreover, in Figure 2 one sample was not marked in the legend. Please also pay attention to the illogical presentation of the series sequence in the charts.

- The TGA graph should be supplemented with a dTG curve and should present GTR data.

- The wording "interfacial cohesion" in line 167 is not correct and should be corrected.

- The crystallinity of PVC mentioned in line 307 is a contentious and debatable issue, I am asking the authors for comment.

Author Response

REVIEWER 2

- In the literature published so far, there are many reports on composites' production with a matrix from various thermoplastic polymers filled with GTR. Many of them also describe the dielectric properties of such copies, defining the final products as potential insulators. Earlier works by the authors also present very similar studies. Considering that the dielectric properties of EPDM are known, the final result of the research is predictable. Therefore, in my opinion, the novelty of the presented work raises doubts.

Answer: The main novelties and contributions in the manuscript work are in dielectric aspects and the terms of dielectric relaxations characterization for EPDM/GTR blends. The interfacial polarization phenomena observed make changes in dielectric behavior, and the Maxwell-Wagner-Sillars (MWS) relaxation checked is very prominent, this is a remarkable aspect of the present research. At the dielectric level, the changes that cause the introduction of GTR in the EPDM matrix, are outstanding highlights, and especially above from 10%GTR amounts in the EPDM composites, like is seen in the dielectric analysis manuscript section. Moreover, is evaluated an electrical viable application for these types of composites.

- Apart from the lack of a significant aspect of novelty, the work constitutes a complete elaboration of the topic and describes the produced materials' properties clearly. The experimental work plan is prepared correctly, and the descriptions of the research results are reliable and well-justified with references.

Answer: Thank the reviewer for the comment.

- The work requires mainly editing corrections related to the presentation of measurement data. Correction of the diagrams (e.g. Fig. 9) should be taken into account by presenting the test results on the same scale and correcting the descriptions under the diagrams. The axes in other drawings and tables must also be corrected by standardizing marking decimal places (I suggest resigning from engineering designations, e.g. E-11) and repeating descriptions under the drawings.

Answer: The work has been edited; axes and decimal places have been standardized. According to the reviewer suggestions, we have adopted the engineering designation for the data tables presentation, and we have taken 2 decimal places in the whole of the manuscript. On the other hand, repeating descriptions underdrawings have been removed.

- All the drawings must be redrawn using the same size and type of fonts. Moreover, in Figure 2 one sample was not marked in the legend. Please also pay attention to the illogical presentation of the series sequence in the charts.

Answer: We have redrawn all the drawings from the manuscript, using the same size and font type. Moreover, in figure 2, the legend has been completed re-edited to clarify the series sequence.

- The TGA graph should be supplemented with a dTG curve and should present GTR data.

Answer: According to the reviewer, we have added a dTG curve, including the GTR data, in Figure 2b. Differences between composites and the GTR are checked from dTG curves analysis, which is supplemented in the manuscript current version (in Figure 2b).

- The wording "interfacial cohesion" in line 167 is not correct and should be corrected.

Anser: According to the reviewer, the authors have modified "interfacial cohesion" from the main text.

- The crystallinity of PVC mentioned in line 307 is a contentious and debatable issue, I am asking the authors for comment.

Answer: The authors have reviewed the revisor comment, and we have detected a mistake, PP is semicrystalline polymer, not PVC. We have changed this (PVC) from the main text.
